# Differences in medication beliefs between pregnant women using medication, or not, for chronic diseases: a cross-sectional, multinational, web-based study

Sonia Roldan Munoz  ,[1] Angela Lupattelli,[2] Sieta T de Vries,[1] Peter G M Mol,[1] Hedvig Nordeng[2]

[1]Clinical Pharmacy and Pharmacology, University Medical Centre Groningen, University of Groningen, Groningen, The Netherlands
[2]Pharmaco Epidemiology and Drug Safety Research Group, Department of Pharmacy, University of Oslo, Oslo, Norway

**Correspondence to**
Professor Hedvig Nordeng;
h.m.e.nordeng@farmasi.uio.no

## ABSTRACT

**Objectives** To assess whether medication beliefs differ between women who use or not use medication for their somatic chronic diseases during pregnancy and whether this association varies across diseases.

**Design** Cross-sectional web-based survey.

**Setting** Multinational study in Europe.

**Participants** Pregnant women or women with children less than 1 year old from European countries and with asthma, allergy, cardiovascular, rheumatic diseases, diabetes, epilepsy and/or inflammatory bowel diseases (IBD).

**Primary and secondary outcome measure** Differences in scores of the *Beliefs about Medicines Questionnaire* (BMQ).

**Results** In total, 1219 women were included (ranging from 736 for allergy to 49 for IBD). Women using medication for their chronic disease (n=770; 63%) had higher scores on the BMQ subscales *necessity* (16.6 vs 12.1, p<0.001) and *benefits* (16.2 vs 15.4, p<0.001), and lower values on the subscales *overuse* (12.5 vs 13.1; p=0.005) and *harm* (9.8 vs 10.7, p<0.001) than women not using medication. No significant differences were shown for the *concerns* subscale (12.5 vs 12.3, p=0.484). Beliefs varied somewhat across diseases but in general more positive beliefs among women using medication were shown. Epilepsy was the disease where less differences were observed between women using and not using medication.

**Conclusion** Women's beliefs were associated with medication use during pregnancy with only small differences across the diseases. Knowing pregnant women's beliefs could help identify women who are reluctant to use medication and could guide counselling to support making well-informed treatment decisions.

### Strengths and limitations of this study

► This study is the first comparing medication beliefs among women using or not using medication during pregnancy for various chronic diseases.
► Over 1000 women from several European countries were included in the study.
► Diseases and medication in this study are based on self-reported data and the recruitment method cannot discard selection bias.

## INTRODUCTION

Worldwide, the number of women using over-the-counter or prescribed medication during pregnancy has been estimated to be 80%–90%.[1 2] Although many prescribed medications during pregnancy are for pregnancy-related symptoms, a substantial number of prescriptions are also to treat chronic diseases.[3] However, adherence to chronic medication during pregnancy is low. Previous studies among pregnant women with various chronic diseases (eg, asthma, epilepsy and inflammatory bowel diseases) have shown that about 40% of these women do not adequately adhere to their medication,[4 5] which challenges appropriate management of the underlying maternal disease.[6] Suboptimally treated maternal chronic diseases like epilepsy, asthma, diabetes and mental disorders can have a negative impact on the mother and on the unborn child (eg, low birth weight, macrosomia, preterm birth or perinatal mortality).[7–11] So, adequate medication use is essential for both maternal and child health.

There are several theoretical models that can be used to explain and improve behaviours such as medication taking. An example is the health belief model in which beliefs are associated with behaviours.[12] Previous studies have shown that patients' medication beliefs are an important factor that can influence treatment adherence.[5 13–15] These beliefs and their association with adherence may differ across diseases.[16] Women generally perceive

medication use during pregnancy as potentially harmful for the unborn child, even for medications that have been proven to be safe.[4 17–19] A newly published literature review concluded that women were more reluctant to use medication during pregnancy and that they tend to overestimate the teratogenic risk of medications. The perceived risk is influenced by different factors including the type of medication.[5] Currently, little is known about pregnant women's medication beliefs and its association with medication use during pregnancy.

The primary aim of this study was to assess whether medication beliefs differ between women who use medication (medicated women) during pregnancy for their somatic chronic diseases (ie, allergy, asthma, cardiovascular diseases, rheumatic diseases, diabetes, epilepsy and inflammatory bowel diseases) and those who do not use such medication (non-medicated women) during pregnancy. The secondary aim was to assess whether differences in beliefs between medicated and non-medicated women vary depending on the disease.

## PARTICIPANTS AND METHODS
### Participants
This is a substudy of the 'Multinational Medication Use in Pregnancy Study', a cross-sectional, multinational web-based survey study performed in 18 countries in Europe, America and Australia. The study has been described in details elsewhere.[2] In short, pregnant women at any gestational week and mothers with a child less than 1 year of age were asked via banners on national websites or social media networks frequently visited by pregnant women and new mothers to complete an anonymous online questionnaire (www.questback.com). This questionnaire was accessible between 1 October 2011 and 29 February 2012. The questionnaire contained a variety of questions including demographics (ie, region of residence, age, marital status, working status, educational level, smoking status before pregnancy and immigrant status), and pregnancy-related questions (ie, maternal status of pregnancy, previous children, use of folic acid during the pregnancy, smoking status and alcohol intake during pregnancy and whether the pregnancy was planned or not). The full questionnaire and the list of websites and social networks used in each country have been published previously.[2]

In this substudy, women from any European country who had at least one somatic chronic disease were included. Women were asked to indicate which chronic diseases they had from a list of diseases: asthma, allergy, rheumatic diseases (including rheumatic arthritis and psoriatic arthritis), diabetes (type I or II), epilepsy and cardiovascular diseases (including high cholesterol, hypertension and heart disease). In addition, the responders could indicate any 'other chronic disease'. Responses provided in this free-text entry field were screened by one of the researchers (AL) and where relevant recoded as one of the diseases of the list. Deep vein thrombosis and

thrombophilia were included as cardiovascular diseases. Also, one additional chronic disease was created based on the open-ended response, that is, inflammatory bowel diseases, consisting of ulcerative colitis and Crohn's disease.

For each chronic disease, women were asked if they were currently using medication and if so, if they could report the medication they use in a free-text entry field. All recorded medications were classified according to the WHO Anatomical Therapeutic Chemical code.[20] Iron preparations, vitamins and minerals were excluded.

The responders were additionally asked the following question: 'Do you have any other comments about your medication use during pregnancy?' The answer to this entry allowed women to further describe their beliefs and perceptions regarding the use of medication during pregnancy. The answers were translated into English and the transcripts were screened to identify general themes by one of the authors (SRM) and discussed with a second author (HN).

### Patient and public involvement
Patients or public were not directly involved in this study.

### Outcome variable
The outcome variable used in this study was pregnant women's medication beliefs, which were assessed using the *Beliefs about Medicines Questionnaire* (BMQ).[21–23] The BMQ was developed by Horne *et al* and contains questions about beliefs about medication in *general* and about *specific* medication. The BMQ-*General* consists of three subscales, that is, *overuse, harm* and *benefits* with four items each. The BMQ-*Specific* consists of two subscales, that is, *necessity* of taking the prescribed medication and *concerns* about the potential adverse consequences of taking the medication. Both scales consist of five items. Each item of the BMQ has to be answered on a 5-point Likert scale, ranging from 'strongly disagree=1' to 'strongly agree=5'. Therefore, the sum scores of the subscales *overuse, harm* and *benefits* range from 4 to 20 and the sum scores of the subscales *necessity* and *concerns* range from 5 to 25, with higher scores representing stronger beliefs of each subscale.

The survey was translated into the official language of the participating countries. Validated versions of the translated BMQ-General and BMQ-Specific subscales were used when they were available.[22 24–29] Otherwise, translation from English and back translations were executed by two independent translators. For all BMQ subscales, missing values were imputed using the estimation-maximisation algorithm if a respondent had no more than two missing items on the subscale (ie, ≥50% completion in the *overuse, harm* and *benefits* subscales; ≥60% completion in the *necessity* and *concern* subscales).[30] If more than two items were missing, the respondent was excluded. BMQ values were ultimately imputed for 68 women, 5.6% of the study population.

Internal consistency was measured calculating Cronbach's alpha for each BMQ subscale per chronic disease.

The lowest and the highest values of Cronbach's alpha were 0.58 (epilepsy) and 0.79 (rheumatic diseases) for *overuse*; 0.62 (epilepsy) and 0.78 (allergy) for *harm*; 0.66 (epilepsy) and 0.82 (asthma) for *benefits*; 0.63 (epilepsy) and 0.85 (diabetes) for *concerns*; and 0.90 (for all diseases) for *necessity*.

Besides the subscales, the *necessity* minus *concerns* differential (*necessity – concerns*) was calculated. A positive differential indicates that the benefits of using medication outweigh the *concerns* whereas the *concerns* weigh higher in case of a negative differential.[31]

### Statistical analyses

Differences between medicated and non-medicated women in their characteristics were analysed using Pearson's $\chi^2$ tests or Fisher's exact tests. T-tests were used to assess differences between medicated and non-medicated women for all included diseases together and per disease in the different BMQ subscales (ie, *overuse, harm, benefits, necessity* and *concerns*) and the *necessity – concerns* differential. In these tests, the included study sample was corrected by survey weight adjustment for age and education level using the reference values from Eurostat.[32] In the weighting procedure, each woman was assigned a weighting factor based on the population proportion per country divided by the sample proportion in each age-by-education stratum per country. Women under-represented in our sample were assigned a weight greater than 1, while those over-represented received a weight smaller than 1.[33] The survey weight for the entire study sample had a mean of 0.93 (range 0.13–5.05). We additionally performed sensitivity analyses in which the t-tests were conducted using the non-weighted study sample. There is discussion in the literature about the use of parametric tests for Likert scales.[34] Therefore, we additionally examined differences in BMQ subscales between medicated and non-medicated women non-parametrically using Wilcoxon-Mann-Whitney tests.

P values <0.05 were considered statistically significant. The statistical analyses were conducted using Stata V.14 (StataCorp, College Station, TX) and figures were created in Microsoft Excel 2010 (Microsoft, Redmond, WA, USA).

### RESULTS

In total, 1219 out of 9483 (12.9%) women reported a somatic chronic disorder and were included in this study. The most prevalent disease was allergy (n=736, 60%), followed by asthma (n=413, 34%), cardiovascular disease (n=238, 20%), rheumatic disease (n=118, 10%), diabetes (n=51, 4.2%), epilepsy (n=50, 4.1%) and inflammatory bowel disease (n=49, 4.0%) (figure 1).

The women were on average 30 years old (SD: 5.2, range: 16–52) and about half of the women (52%) were pregnant at the time of completing the survey (table 1).

About two-thirds of the women reported the use of at least one medication to treat their somatic chronic diseases during pregnancy (n=770, 63%). This ranges

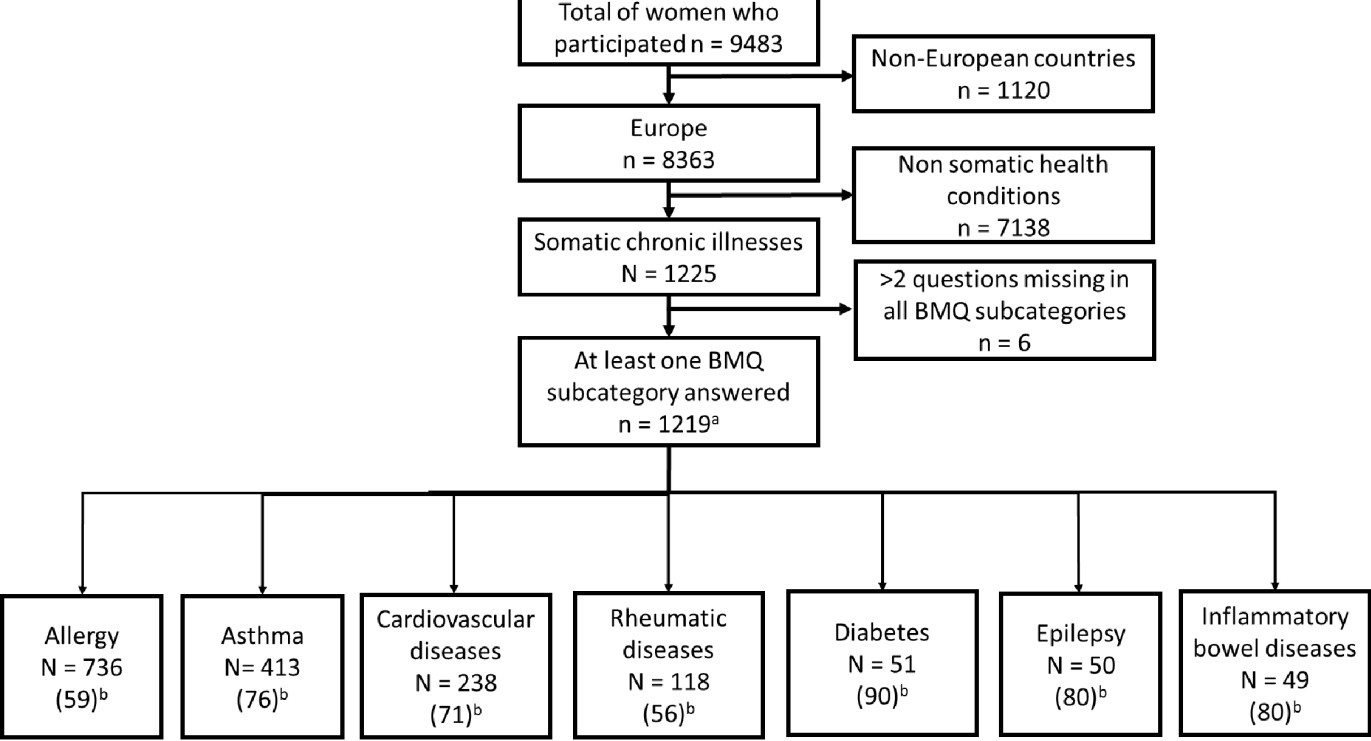

ᵃ372 women reported 2 somatic chronic diseases, 29 women reported 3 and 2 women reported 4.
ᵇ(Percentage of Medicated women)

**Figure 1** Flow chart of the included population. BMQ, Beliefs about Medicines Questionnaire.

**Table 1** Demographics of the included population according to medication use status for the examined chronic diseases during pregnancy

| Maternal characteristics | Total (n=1219) | Medicated (n=770) | Non-medicated (n=449) | P value |
|---|---|---|---|---|
| Region of residence, n (%) | | | | 0.051 |
| Northern Europe* | 552 (45) | 349 (45) | 203 (46) | |
| Western Europe† | 298 (24) | 248 (32) | 121 (27) | |
| Eastern Europe‡ | 369 (30) | 173 (22) | 125 (28) | |
| Maternal age | | | | **0.039** |
| Mean, SD | 30.1, 5.2 | 30.4, 5.1 | 29.5, 5.4 | |
| Range min-max | 16–52 | 16–52 | 17–50 | |
| Currently pregnant | | | | 0.630 |
| Yes | 630 (52) | 402 (52) | 228 (51) | |
| No | 589 (48) | 368 (48) | 221 (49) | |
| Previous children | | | | **<0.001** |
| Yes | 600 (49) | 424 (55) | 195 (43) | |
| No | 619 (51) | 346 (45) | 254 (57) | |
| Marital status | | | | **0.001** |
| Married or cohabiting | 1141 (94) | 734 (95) | 407 (91) | |
| Divorced/single or others | 78 (6) | 36 (5) | 42 (9) | |
| Folic acid use§ | | | | 0.239 |
| Yes | 1102 (91) | 704 (92) | 398 (90) | |
| No | 105 (9) | 61 (8) | 44 (10) | |
| Working status | | | | **0.002** |
| Student | 116 (10) | 59 (8) | 57 (13) | |
| Home maker | 95 (8) | 68 (9) | 27 (6) | |
| Healthcare personnel (ie, physician) | 177 (15) | 125 (16) | 52 (12) | |
| Employed in another sector | 712 (58) | 452 (59) | 260 (58) | |
| Job seeker | 52 (4) | 27 (4) | 25 (6) | |
| None of the above | 66 (5) | 38 (5) | 28 (6) | |
| Educational level | | | | 0.637 |
| Primary school | 66 (5) | 38 (5) | 28 (6) | |
| High school | 338 (28) | 216 (28) | 122 (27) | |
| University or college | 672 (55) | 421 (55) | 251 (56) | |
| Other education | 143 (12) | 95 (12) | 48 (11) | |
| Alcohol use after awareness of being pregnant | | | | 0.354 |
| No | 189 (16) | 641 (84) | 383 (86) | |
| Yes | 1024 (84) | 125 (16) | 64 (14) | |
| Smoking before pregnancy | | | | **0.014** |
| No | 786 (65) | 516 (67) | 270 (60) | |
| Yes | 432 (35) | 253 (33) | 179 (40) | |
| Smoking during pregnancy | | | | 0.631 |
| No | 1105 (91) | 700 (91) | 405 (90) | |
| Yes | 113 (9) | 69 (9) | 44 (10) | |
| Planned pregnancy | | | | 0.495 |
| No | 108 (9) | 65 (8) | 43 (10) | |
| Yes | 1108 (91) | 703 (92) | 404 (90) | |

Continued

**Table 1** Continued

| Maternal characteristics | Total (n=1219) | Medicated (n=770) | Non-medicated (n=449) | P value |
|---|---|---|---|---|
| Immigrant status¶ | | | | 0.138 |
| No | 1176 (97) | 748 (97) | 428 (96) | |
| Yes | 42 (3) | 22 (3) | 20 (5) | |

Numbers may not add up to total due to missing values. Minor missing value is 0.08% and major missing value is 1.31%. Missing values: maternal age n=16, folic acid use n=12, working status n=1, alcohol use after awareness of being pregnant n=6 (cannot remember what is considered as missing n=5), smoking before pregnancy n=1, smoking during pregnancy n=1, planned pregnancy n=4, immigrant status n=1. Statistically significant results (ie, p<0.05) are presented in bold.
Examined chronic diseases include: allergy, asthma, cardiovascular diseases, rheumatic diseases, diabetes, epilepsy and inflammatory bowel diseases.
*Northern Europe includes Finland (13%), Iceland (1.3%), Norway (19%) and Sweden (12%).
†Western Europe includes Austria (0.9%), France (4.2%), Italy (7.1%), Netherlands (1.7%), Switzerland (6.2%) and UK (10%).
‡Eastern Europe includes Croatia (2.6%), Poland (6.9%), Russia (11%), Serbia (1.8%) and Slovenia (1.7%).
§Indicates folic acid use before and/or during pregnancy.
¶Women having the first language different from the official main language in the country of residency.

from 56% of women using medication for rheumatic diseases to 90% of women using medication for diabetes (figure 1). The most commonly reported medications were piperazine derivatives (R06AE) for allergy, selective beta-2-adrenoreceptor agonists (R03AC) for asthma, methyldopa (C02AB) for cardiovascular diseases, glucocorticoids (H02AB) for rheumatic diseases, fast-acting insulin (A10AB) for diabetes, other antiepileptic drugs (N03AX) for epilepsy, and aminosalicylic acid and similar agents (A07EC) for inflammatory bowel diseases.

Some patient characteristics differed significantly between the medicated and non-medicated women. Medicated women were on average older (30.4 vs 29.5 years, p=0.039), more often had previous children (55% vs 43%, p<0.001), were more often married or cohabiting (95% vs 91%, p<0.001) and were less often smokers before knowing about their pregnancy (33% vs 40%, p=0.014). In addition, there were differences in the working status between the medicated and non-medicated women (p=0.002) (table 1).

There were 359 women out of the 1219 who provided an open-ended answer to the question 'Do you have any other comments about your medication use during pregnancy?' From those answers, five themes were identified, that is, confident, underinformed, confused, guilt and avoiding medication. In all themes, there were women from the medicated and non-medicated groups. Examples of statements supporting these themes are presented in table 2.

## Beliefs about medicines

In general, medicated women had more positive beliefs than non-medicated women. Statistically significant differences between the weighted mean scores of the medicated and non-medicated women were shown in the *overuse, harm, benefits* and *necessity* subscales, but not *concerns* subscale. Mean scores of the medicated group were lower than those of the non-medicated group for *overuse* (12.5 vs 13.1, respectively, p=0.005) and *harm*

(9.8 vs 10.7, respectively, p<0.001) and higher for *benefits* (16.2 vs 15.4, p<0.001) and *necessity* (16.6 vs 12.1, p<0.001, respectively). There was no difference on the *concerns* subscale between medicated and non-medicated women (12.5 vs 12.3, p=0.484) (figure 2A). For each disease, statistically significant differences between medicated and non-medicated women were shown in at least two of the subscales except for epilepsy where no statistically significant differences between medicated and non-medicated women were shown for any of the subscales (figure 2B–H).

The *necessity – concerns* differential was positive for all chronic diseases together in the medicated group but it was negative for the non-medicated group. This means that medicated women perceived that the benefits of taking prescribed medication outweigh their risks. The mean difference of the *necessity – concerns* differential varied across the somatic diseases for medicated and non-medicated women. In the medicated groups, the differential was positive for each of the chronic diseases. Women with diabetes had the highest beliefs that the benefits of using medication for their chronic disease outweigh the risks (+9.0), while women with rheumatic diseases had the lowest beliefs in a positive benefit-risk balance (+3.3). For the non-medicated group, the differential for each chronic disease was close to zero, with the highest differential for diabetes (+1.6) and the lowest differential for cardiovascular diseases (−0.6). The differential was statistically significantly higher for medicated than for non-medicated women across all examined chronic disease groups except for pregnant women with epilepsy (figure 3).

The sensitivity analyses in which no survey weight adjustment was used showed similar results to the analyses with survey weight adjustment (see online supplementary materials e-Figure 1 and e-Figure 2). The additional analyses assessing the association between medicated and non-medicated women and the BMQ subscales using a

**Table 2** Identified themes and examples of statements provided by the responders regarding medication use during pregnancy

| Confident | '*I have been to the specialist and I have received good information and support.*'<br>(Medicated woman with inflammatory bowel disease, 30 years old from Norway)<br><br>'*People told me things about amoxicillin that my doctor gave me, but he told me that it is not harmful and that I needed it. Listen to your doctor and trust him.*'<br>(Medicated woman with rheumatic disease, 43 years old from Croatia)<br><br>'*I cannot say that I restricted myself. I used medications as it was needed, I do not think it caused any harm to my child.*'<br>(Non-medicated woman with cardiovascular disease, 23 years old from Russia) |
| --- | --- |
| Underinformed | '*Which medicines are proven harmless to the foetus, percentages, thank you! Many are afraid to take any medicines during pregnancy because of an exact answer may not be.*'<br>(Non-medicated woman with cardiovascular disease, 26 years old from Finland)<br><br>'*It is very difficult to find a drug that is acceptable in pregnancy. Even for very simple things it is stated that pregnancy is a contraindication.*'<br>(Non-medicated woman with cardiovascular disease, 26 years old from Russia)<br><br>'*I chose to use medication minimally due to uncertain information about how this could affect the foetus.*'<br>(Non-medicated woman with cardiovascular disease, 26 years old from Norway) |
| Confused | '*Frequently we see differences in drugs information and in recommendations from our general doctors or gynaecologists. Should we believe the drug information or to the other side?*'<br>(Medicated woman with asthma and allergy, 31 years old from Croatia)<br><br>'*I find it hard to be in the middle of opinions from doctors, between the gynaecologist and the medical specialist.*<br>*One says I can take all my medication, while the gynaecologist said that everything must be stopped. Who to believe?*'<br>(Medicated woman with inflammatory bowel disease and rheumatic disease, 26 years old from Switzerland)<br><br>'*To me it seems like that there is a kind of psychological terror on the use of medications. To some doctors, no medication can be taken during pregnancy, others give far too many possibilities in this regard.*'<br>(Medicated woman with rheumatic disease, 39 years old from Italy) |
| Guilt | '*In a previous pregnancy, I used Panadol every day to painful knee ache, so that I was able to walk. So my son was born with a multicystic kidney and an undescended testicle. I think that the drug could have an impact on birth defect.*'<br>(Non-medicated woman with inflammatory bowel disease, 24 years old from Finland)<br><br>'*Unfortunately, I had to take many medications and I felt very guilty (and I still do feel guilty) but if I did not take them, my child would probably not be born.*'<br>(Medicated woman with cardiovascular disease, 37 years old from Italy)<br><br>'*I've reduced after consultation with the neurologist the dosage, as I had concerns for the child. It came to seizures because the level was too low, so the dose had to be increased again. Despite my concerns, I had to take the medication. It was very difficult for me, but I had no choice.*'<br>(Medicated woman with epilepsy, 30 years old from Switzerland) |
| Avoiding medication | '*I stopped taking my medication as soon as I knew I was pregnant.*'<br>(Medicated woman with rheumatic disease, 27 years old from Switzerland)<br><br>'*Generally: I'm terrified to hurt my foetus and try to avoid all drugs, both with and without a prescription. I have sleeping problems, but do not take the pill the doctor has prescribed to me (for the same reason). But daily life becomes heavy when you cannot sleep.*'<br>(Non-medicated woman with rheumatic disease, 36 years old from Norway)<br><br>'*I intentionally did not take medications during pregnancy.*'<br>(Non-medicated woman with asthma and allergy, 26 years all from Russia) |

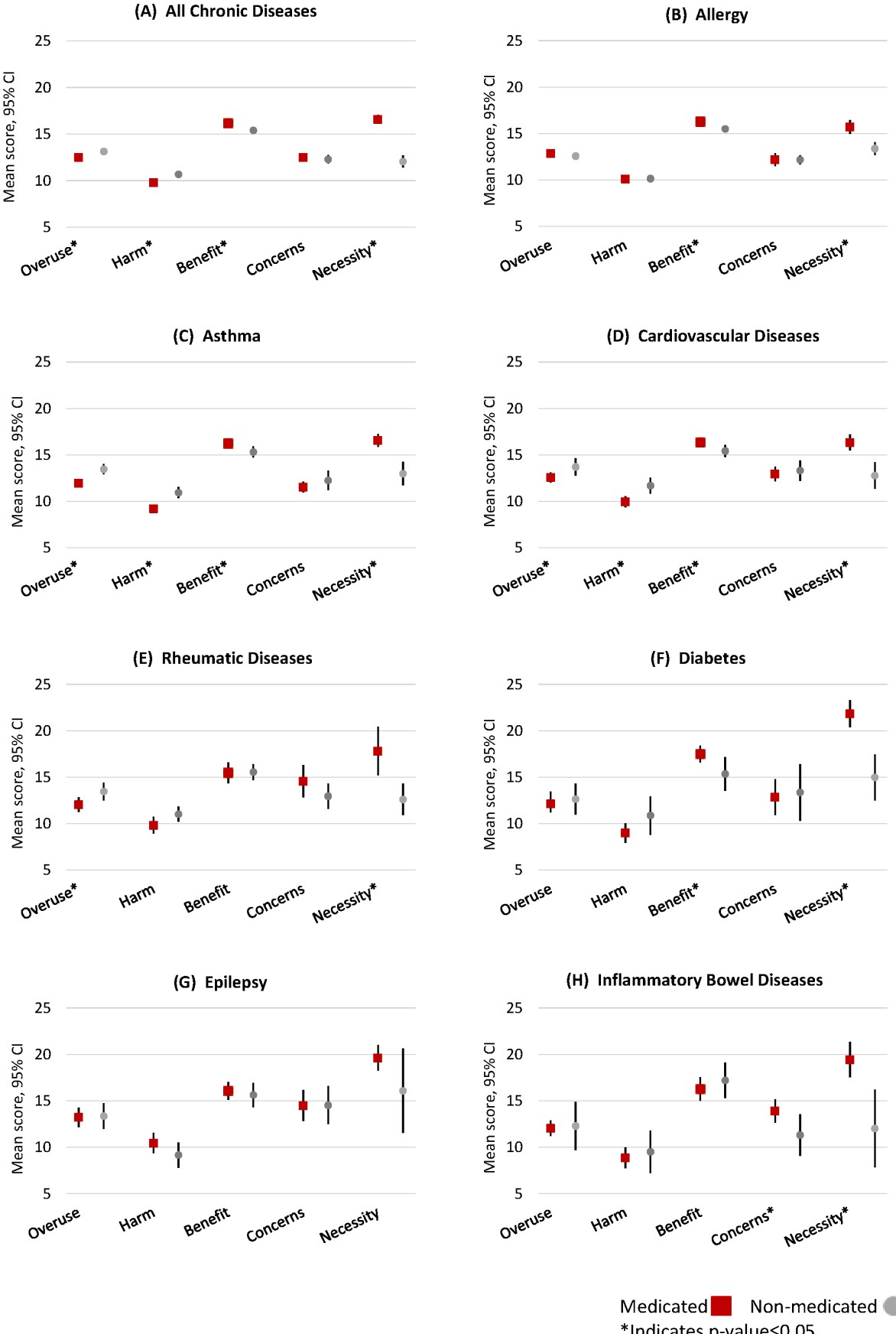

Medicated ■  Non-medicated ●
*Indicates p-value≤0.05

**Figure 2** *Beliefs about Medicines Questionnaire* (BMQ) weighted mean score with 95% CIs for (A) each BMQ subscale for all chronic diseases and for (B-H) each chronic disease by use of medication. The BMQ-Specific and BMQ-General are copyrighted (Professor Robert Horne).

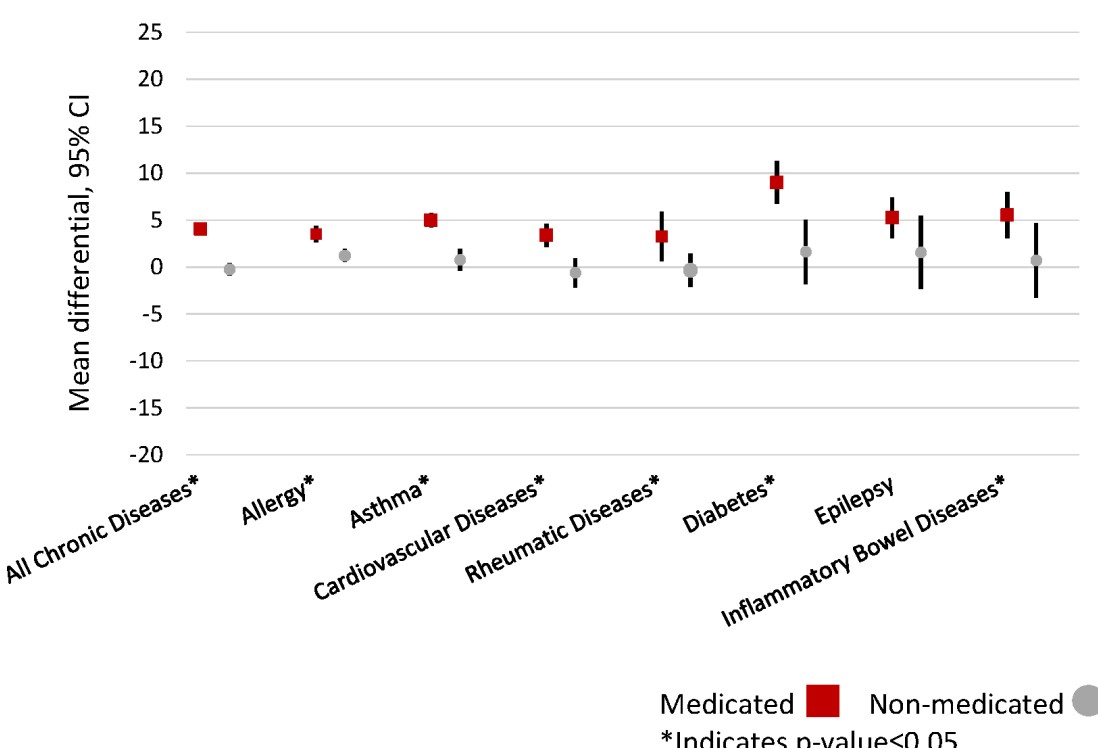

**Figure 3** *Beliefs about Medicines Questionnaire* (BMQ) weighted *necessity – concerns* differential for all chronic diseases and for each chronic disease by use of medication. Values higher than 1 indicate that the necessity of the medications is larger than the concerns. The BMQ-Specific and BMQ-General are copyrighted (Professor Robert Horne).

non-parametric approach showed similar results (see online supplementary materials e-Table 1).

## DISCUSSION

In our study, more than a third of the responding women did not use medication for their chronic disease during pregnancy. These women had more negative beliefs than the two-thirds of women who used or continued to use medication during pregnancy. This difference was shown across different chronic diseases with the exception of epilepsy. For epilepsy, medicated and non-medicated pregnant women had similar medication beliefs.

The proportion of women using medication for their somatic chronic disease varied widely across the diseases in our study (from 56% for rheumatic diseases to 90% for diabetes). This variation in medication use during pregnancy is in line with previous studies showing percentages ranging from 62% for the treatment of chronic psychiatric disorders[35] to 93% for the treatment of hypothyroidism.[36] Another study showed that patient characteristics such as educational level and occupation influence pregnant women's perceived risks of medication use.[5] We also observed some differences in sociodemographic characteristics between medicated and non-medicated women, such as maternal age, marital status or having previous children. Further studies should asses the role of these

factors in the association between beliefs and the decision to use medication.

Previous studies in pregnant women have shown that positive medication beliefs are associated with higher medication use and better treatment adherence.[31 35 36] Our study supports this association from a broader medication-taking perspective, and adds to this knowledge that the differences in beliefs between medicated and non-medicated pregnant women were shown across various diseases, except for epilepsy. For epilepsy, medicated and non-medicated women had similar beliefs with particularly high *concerns* beliefs. Previous studies have shown that more than 50% of epileptic women would discontinue their medication during pregnancy.[37 38] It is long understood that some antiepileptics (ie, valproic acid and phenytoin) are teratogenic and more recent findings also indicate neurodevelopmental risks to the offspring.[39] This may explain the similarity in concerns about medication among medicated and non-medicated women with epilepsy.

The provided comments to the free-text entry outline the lack of information among patients and healthcare professionals on the adequate use of medication during pregnancy. This is highly relevant since knowledge is a prerequisite for the appropriate use of a treatment among pregnant women. However, information about medication use during pregnancy is often lacking. A

previous study showed, for instance, that 90% of the reviewed summary of product characteristics (SmPC) were restricting the use of medication during pregnancy but that SmPCs often do not provide a clear rationale for this restricted use. More specifically, 89% of those SmPCs did not specify whether or not the medication crossed the placenta and in 67% of the SmPCs it was just stated that clinical experience data were not available.[40] Moreover, SmPCs do not provide alternative solutions. Therefore, the various healthcare professionals involved in the treatment of pregnant women (eg, general practitioners, specialists treating their chronic diseases, pharmacists, midwives and gynaecologists) will have to rely on other, sometimes scarce or diverging information sources which could result in inconsistent advice to pregnant women. Therefore, more and clearer information about the use of medication during pregnancy is needed.

Not using medication for a chronic disease does not necessarily imply inappropriate behaviour of the pregnant women since the need of medication use depends on factors such as disease severity and other external factors (eg, for allergy, there is a low risk of events in certain seasons of the year). However, positive medication beliefs among women could contribute to the decision of continuing their medication during pregnancy. Therefore, patients' beliefs should be discussed with the woman, and this information can be used to target those patients who are reluctant to continue with the required medication during pregnancy. Educational interventions have been successful in increasing appropriate medication use in pregnant and non-pregnant patients in various situations as well as on attenuating negative beliefs and enhancing positive beliefs.[41–44] This could lead to better disease control. Innovative decision support tools that empower women and enable a shared decision-making approach about medications in pregnancy have been developed for some diseases[45] and could lead to better informed treatment decisions. Further development of such tools, also for other diseases, should be the focus of the future. If a patient still decides not to continue a treatment during pregnancy, closer monitoring to the maternal and child health should be conducted.

### Strengths and limitations

To our knowledge, this is the first study in which medication beliefs are compared between pregnant women who use and who do not use medication for their somatic chronic disease. Moreover, we assessed these differences for various diseases in a large sample of women from several European countries.

Although a large number of women were included in this study, the numbers for some diseases are relatively low, particularly for those non-medicated, and the results per disease should therefore be interpreted cautiously.

Another limitation of this study is that the diseases and used medications are based on self-reporting without a medical confirmation. This implies that the accuracy of the data depended on the accuracy of the women.

However, a previous study comparing self-reported and pharmacy records showed sufficient accuracy of self-reported medication use.[46]

It is known that survey responders have the tendency to respond relatively more neutral to Likert scale questions.[47] Therefore, we have examined the distribution of the answers for each BMQ question for each disease in our study and we observed large variability in the distributions (see online supplementary material e-Table 2.)

We used the BMQ to assess pregnant women's beliefs about medication. However, also other beliefs could be relevant to assess in this population. Future research should develop validated measures of pregnancy-specific beliefs about prescribed and over-the-counter medicines, and use patient-centred approaches in the development of the items.

Furthermore, due to the recruitment method used in this study it was not possible to calculate response rates and selection bias cannot be discarded. The educational level in our sample was higher than in the average population per country. To take this into account, we employed sample survey weights. The reference values used in this weighting represented women in reproductive age in each of the European countries[32] but were not specific of the pregnant population.

### CONCLUSION

In this multinational European study we found an association between women's beliefs and medication use in pregnancy. Women with somatic chronic diseases using medication in pregnancy had more positive medication beliefs than those who did not use medication. This result was independent of the underlying disease, as the association was shown in each disease except for epilepsy, where medicated and non-medicated women had similar medication beliefs. In addition, it was shown that women with chronic diseases lacked confidence, felt underinformed, confused, guilty about medication use and avoided them during pregnancy. Appropriate educational interventions and innovative decision support tools to assist tailored counselling and guidance before and during pregnancy are therefore warranted, and could lead to more positive beliefs, enhancing a well-informed decision about medication use during pregnancy.

**Acknowledgements** We thank the Scientific Board of the Organization of Teratology Information Specialists (OTIS) and the European Network of Teratology Information Services (ENTIS), the website providers who contributed to the recruitment phase, the national coordinators of the study and all women who participated in this study. We thank Professor Robert Horne for giving us permission to use the BMQ-General and BMQ-Specific in the 'Multination Medication Use In Pregnancy Study'.

**Contributors** AL and HN collected the data. All authors contributed to the formulation of the research question of this study. SRM, AL and STdV analysed the data. All authors contributed to the interpretation of the data. SRM drafted the manuscript. AL, HN, STdV and PGMM reviewed and edited the manuscript. All authors read and approved the final version of the manuscript.

**Funding** The study was partly conducted in the context of the PROMINENT project. This project has received funding from the European Union's Horizon 2020 research and innovation programme under the Marie Skłodowska-Curie grant agreement number 754425. The study has also received support from the Foundation for Promotion of Norwegian Pharmacies and the Norwegian Pharmaceutical Society, Oslo, Norway. AL's postdoctoral research fellowship was funded through the HN's European Research Council Starting Grant 'DrugsInPregnancy' (grant number 639377).

**Disclaimer** The funding sources had no involvement in any of the stages from the study design to the submission of the paper for publication.

**Competing interests** None declared.

**Patient consent for publication** Not required.

**Provenance and peer review** Not commissioned; externally peer reviewed.

**Data availability statement** For this study, anonymous data from The Multinational Medication Use in Pregnancy Study are used. The data are available for researchers upon request by contacting the corresponding author.

**ORCID iD**
Sonia Roldan Munoz http://orcid.org/0000-0003-1471-9713

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
