## [Reviewer comments · BMJ Open]

ARTICLE DETAILS

TITLE (PROVISIONAL)	Differences in medication beliefs between pregnant women using medication, or not, for chronic diseases: a cross-sectional, multinational, web-based study.
AUTHORS	Roldan Munoz, Sonia; Lupattelli, Angela; de Vries, Sieta T; Mol, Peter GM; Nordeng, Hedvig

VERSION 1 – REVIEW

REVIEWER	Molly M. Lynch RTI International U.S.A.
REVIEW RETURNED	24-Oct-2019

GENERAL COMMENTS	I enjoyed reading this paper and felt it contributes to the field. A couple comments:  1. the open ended responses seem critical to the discussion of results. It is somewhat a foregone conclusion that beliefs contribute to behaviors, so a more thorough discussion of open-ended coding in the method would be helpful, as well as additional discussion in the results and conclusions. What more can you say about the why of the beliefs that contributes to actual practice implications? 2. Because we know that beliefs contribute to behavioral intentions, framing results in the form of a theoretical model would add strength to the results of this paper. For example, the Health Belief Model or Theory of Reasoned Action tell us that beliefs determine behaviors. I recommend placing the results into one of these frameworks, and discussing the other constructs that might influence behaviors (e.g., cue to action in the Health Belief Model).
---

REVIEWER	Jan Schjøtt Department of Medical Biochemistry and Pharmacology, Haukeland University Hospital, Bergen, Norway Department of Clinical Science, Faculty of Medicine and Dentistry, University of Bergen, Bergen, Norway
REVIEW RETURNED	18-Nov-2019

GENERAL COMMENTS	Review Knowledge of medication beliefs among pregnant with a chronic disease is important for planning drug information efforts. The authors have gathered a large multinational material through a web-based survey. The study shows that drug information efforts is needed among women with chronic diseases, and indicate that a significant number of women are perhaps reluctant to use medications during
--

pregnancy. This is important to communicate to the health care system.

General comments

The main results concerning beliefs about medicines are not surprising, and perhaps expected by the authors. Medicated women were older (and perhaps more experienced) with previous children (healthy in spite of medication), and they had more frequently a spouse or cohabitat to discuss any concern associated with medications with. Furthermore, severity of the chronic disease (not measured in this study) could have influenced the results due to shared expectations and perceptions between pregnant and health care professionals. The results of the open-ended questions give important additional information about perception among the women. The overall results can be used to raise hypotheses to be tested in subsequent studies among women with chronic diseases. In particular with regard to type of chronic disease and/or type of medication. The manuscript is concise and well written with well structured figures and tables. I suggest a minor revision based mainly on some methodological comments:

Specific comments

I am no expert on the BMQ instrument. Thus, I have some questions/comments that could reflect the average reader of BMJ Open. The questions/comments could be addressed briefly by the authors to potentially improve the manuscript:

a) T-tests were used to assess differences

between medicated and non-medicated women for all included diseases together and per disease in the different BMQ subscales. Is it appropriate to use T-test when the basis of the BMQ is a Likert scale with items consisting of ordinal categories (see also below)? See: <https://www.sim-one.ca/community/tip/analyzing-likert-scale-data-rule-n30>

b) Concerning the use of t-test and comparing means. We do not know if its equal distance between the individual categories in the Likert scales. So although we find a significant difference between a sum score for the respective groups, a large proportion of the scores could be close to the more neutral choices in the Likert scale. Was it so? With very large sample sizes, statistical power can be so high that impractically small changes (effects) are statistically significant but not of meaningful (practical) importance. The sum scores are between 10-15 in Figure 2 A-H. Does this mean that a significant percent of medicated or non-medicated scored in the neutral category in some or several of the items of BMQ?? Notably, some authors describe percent neutral scores in addition to means.

c) How good is the instrument across different chronic diseases?

Could disease-or condition specific instruments be more useful?

Pharmacotherapy (e.g. formulations, administration, monitoring, empowerment, etc.), quality of life, empowerment is so different between chronic diseases that this could be of relevance. Diabetes is an example where the patient becomes an expert on his or hers disease. This is perhaps not the case with some of the other diseases. See:

<https://www.ncbi.nlm.nih.gov/pmc/articles/PMC5556958/> or

<https://bmchealthservres.biomedcentral.com/articles/10.1186/s12913-017-2020-y>

d) How good is the test instrument in the context of a multinational survey across several countries? Is medicines costs and reimbursement across the included countries that participated very different? Is the treatment tradition, role and function of the health care system very different among the participating countries? In the context of people using regular medicines, is BMQ adequate? See:

	https://www.ncbi.nlm.nih.gov/pmc/articles/PMC5063133/. Should we use patients to generate items in subsequent studies? How important is treatment burden with a chronic disease? Does it influence beliefs about medicines? Is there any language barriers? Did all participants use an english or a translated version of BMQ? e) It has been long acknowledged that the extremes of a Likert-type response tend to get less use than the more central choices causing an “anchor effect”. Could this be a potential explanation for the results beeing independent of the underlying disease (except epilepsy where documentation for harm of medications is very strong)? Therefore, the intervals near the extremes may be further apart, than those near the center. This, by itself, disqualifies a Likert-type response as interval. See: https://www.ncbi.nlm.nih.gov/pmc/articles/PMC4833473/ f) The authors state in the paragraph starting at line 53 on page 11 that this study supports treatment adherence, although adherence was not specifically adressed in the survey. I guess that BMQ scores (and in particular necessity-concerns) are used as a proxy for adherence in studies. g) How many open-ended answers were used to create the five themes? Should this number be included in Results, and the respective text below Table 2? Finally, due to the above mentioned challenges with measuring different chronic diseases, pregnant and beliefs of medications, more studies are needed to find the appropriate and more specific interventions. This notion should preferently be added to the conclusion. Thanks for this opportunity to review an interesting study.
--	---

VERSION 1 – AUTHOR RESPONSE

Reviewer: 1

1. the open ended responses seem critical to the discussion of results. It is somewhat a foregone conclusion that beliefs contribute to behaviors, so a more thorough discussion of open-ended coding in the method would be helpful, as well as additional discussion in the results and conclusions. What more can you say about the why of the beliefs that contributes to actual practice implications?

Reply 1. We agree that the open-ended answers provide additional insights about women beliefs and perceptions on medication use. These responses are of high value, however, we consider these results explorative since it is only one open question and not all participants responded this question. Based on your comment as well as the comment of reviewer 2, we have added the number of responders to this question to the Results section (page 9).

The paragraph now reads: *“There were 359 women out of the 1219 who provided an open-ended answer to the question ‘Do you have any other comments about your medication use during pregnancy?’. From those answers, five themes were identified, that is confident, under-informed, confused, guilt, and avoiding medication. In all themes, there were women from the medicated and non-medicated groups. Examples of statements supporting these themes are presented Table 2.”*

The provided answers to the free text entry outline the lack of information among patients and healthcare professionals on the adequate use of medication during pregnancy. This is important

information since it is related to appropriate treatment use and it is mentioned in the Discussion (page 12).

The paragraph now reads: *'The provided comments to the free text entry outline the lack of information among patients and healthcare professionals on the adequate use of medication during pregnancy. This is highly relevant since knowledge is a prerequisites for the appropriate use of a treatment among pregnant women. However, information about medication use during pregnancy is often lacking.'*

2. *Because we know that beliefs contribute to behavioral intentions, framing results in the form of a theoretical model would add strength to the results of this paper. For example, the Health Belief Model or Theory of Reasoned Action tell us that beliefs determine behaviors. I recommend placing the results into one of these frameworks, and discussing the other constructs that might influence behaviors (e.g., cue to action in the Health Belief Model).*

Reply 2. Beliefs are, indeed, one of the aspects that may influence behaviour. The open-ended question answers show that other aspects (e.g. knowledge) might also influence behaviour. As mentioned by the reviewer, this is supported by theoretical models. We have added this in the Introduction section (page 4)

The Introduction now reads: *'There are several theoretical models that can be used to explain and improve behaviours such as medication taking. An example is the Health Belief Model in which beliefs are associated with behaviours¹². Previous studies have shown that patients' medication beliefs are an important factor that can influence treatment adherence^{5, 13-15}.*

Reviewer: 2

a) *T-tests were used to assess differences between medicated and non-medicated women for all included diseases together and per disease in the different BMQ subscales. Is it appropriate to use T-test when the basis of the BMQ is a Likert scale with items consisting of ordinal categories (see also below)?*

Reply a. We agree with the reviewer that there is discussion about the appropriateness of using parametric tests for Likert scales especially since assumptions might be violated. However, there is no need to directly discard the use of a parametric approach (<https://www.sciencedirect.com/science/article/pii/S1877129715200196>) and T-tests are generally quite robust statistics. Given the discussion, we, however, also conducted the analyses in a non-parametric approach (using Wilcoxon-Mann Whitney tests). This was done in the non-weighted sample of our study and the analyses did not show large differences. We have added the results of both the parametric and non-parametric tests as supplementary material (e-Table 1.) and we have amended the Methods (page 8) and the Results (page 10) sections accordingly.

The Methods section was amended as follows: *"There is discussion in the literature about the use of parametric tests for Likert scales³⁴. Therefore, we additionally examined differences in BMQ subscales between medicated and non-medicated women non-parametrically using Wilcoxon-Mann Whitney tests."*

The Results section was amended as follows: *'The additional analyses assessing the association between medicated and non-medicated women and the BMQ subscales using a non-parametric approach showed similar results (see supplementary material, e-Table 1.)'*

b) Concerning the use of t-test and comparing means. We do not know if its equal distance between the individual categories in the Likert scales. So although we find a significant difference between a sum score for the respective groups, a large proportion of the scores could be close to the more neutral choices in the Likert scale. Was it so? With very large sample sizes, statistical power can be so high that impractically small changes (effects) are statistically significant but not of meaningful (practical) importance. The sum scores are between 10-15 in Figure 2 A-H. Does this mean that a significant percent of medicated or non-medicated scored in the neutral category in some or several of the items of BMQ?? Notably, some authors describe percent neutral scores in addition to means.

Reply b. Thank you for this comment. The first part of this question relates to the use of parametric and non-parametric tests (see our response in point 'a)'). Furthermore, it is indeed known that responders may tend to stay in the more neutral scores in questionnaires. To see to what extent this also occurred in our study, we have extracted the percentage of responders answering each answer category of each BMQ question and for each disease. From this table (e-Table 2.) it becomes clear that for several questions, the more extreme scores are selected by a higher percentage of women than the neutral scores. The percentages strongly vary depending on the question. This suggests that the findings reflect actual beliefs instead of an anchor effect.

We have added this table as supplementary material and we have amended the Limitations section (page 13) as follows: "It is known that survey responders have the tendency to respond relatively more neutral to Likert scale questions⁴⁷. Therefore, we have examined the distribution of the answers for each BMQ question for each disease in our study and we observed large variability in the distributions (see supplementary material, e-Table 2.)"

c) How good is the instrument across different chronic diseases? Could disease-or condition specific instruments be more useful? Pharmacotherapy (e.g. formulations, administration, monitoring, empowerment, etc.), quality of life, empowerment is so different between chronic diseases that this could be of relevance. Diabetes is an example where the patient becomes an expert on his or hers disease. This is perhaps not the case with some of the other diseases. See: <https://www.ncbi.nlm.nih.gov/pmc/articles/PMC5556958/> or <https://bmchealthservres.biomedcentral.com/articles/10.1186/s12913-017-2020-y>

Reply c. In previous studies, the BMQ has been used for various diseases (e.g. asthma, diabetes, IBD, cardiovascular diseases (<https://sciforschenonline.org/journals/drug/JDRD-3-130.php>). A meta-analyses shows that the BMQ has been applied in studies assessing more than 24 different diseases (<https://www.ncbi.nlm.nih.gov/pmc/articles/PMC3846635/pdf/pone.0080633.pdf>). In addition, patients with various chronic diseases were included in the development of the BMQ, in order to obtain a general questionnaire of use in different chronic diseases (<https://www.tandfonline.com/doi/pdf/10.1080/08870449908407311?needAccess=true>). Therefore, we consider the use of the BMQ across different diseases a valid approach.

To support our findings, we have also added a paragraph in the Methods section with the lowest and highest alpha values for each BMQ-subscale (page 7).

The paragraph reads as follows: "Internal consistency was measured calculating Cronbach's alpha for each BMQ-subscale per chronic disease. The lowest and the highest value of Cronbach's alpha were are 0.58 (epilepsy) and 0.79 (rheumatic diseases) for overuse; 0.62 (epilepsy) and 0.78 (allergy) for harm; 0.66 (epilepsy) and 0.82 (asthma) for benefits; 0.63 (epilepsy) and 0.85 (diabetes) for concerns; and 0.90 (for all diseases) for necessity."

See in the table below the complete overview of alpha values for each BMQ-subscale per disease. If it is considered of value by the reviewers or editor, we can add this table as supplementary material and refer to it in the manuscript.

BMQ specific

Necessity		Concerns	
All Chronic diseases	0.9028	All Chronic diseases	0.7624
Allergy	0.9029	Allergy	0.7442
Asthma	0.9030	Asthma	0.7660
Cardiovascular diseases	0.9031	Cardiovascular diseases	0.7372
Rheumatic diseases	0.9032	Rheumatic diseases	0.7353
Diabetes	0.9033	Diabetes	0.8503
Epilepsy	0.9034	Epilepsy	0.6314
IBD	0.9035	IBD	0.7131

BMQ general

Overuse		Harm		Benefits	
All Chronic diseases	0.7217	All Chronic diseases	0.7706	All Chronic diseases	0.8031
Allergy	0.6924	Allergy	0.7846	Allergy	0.8098
Asthma	0.7011	Asthma	0.7780	Asthma	0.8179
Cardiovascular diseases	0.7285	Cardiovascular diseases	0.7814	Cardiovascular diseases	0.7562
Rheumatic diseases	0.7948	Rheumatic diseases	0.7736	Rheumatic diseases	0.8068
Diabetes	0.6487	Diabetes	0.7469	Diabetes	0.7523
Epilepsy	0.5835	Epilepsy	0.6267	Epilepsy	0.6603
IBD	0.7469	IBD	0.7437	IBD	0.8060

d) How good is the test instrument in the context of a multinational survey across several countries? Is medicines costs and reimbursement across the included countries that participated very different? Is the treatment tradition, role and function of the health care system very different among the participating countries? In the context of people using regular medicines, is BMQ adequate? See: <https://www.ncbi.nlm.nih.gov/pmc/articles/PMC5063133/>.

Should we use patients to generate items in subsequent studies? How important is treatment burden with a chronic disease? Does it influence beliefs about medicines? Is there any language barriers? Did all participants use an english or a translated version of BMQ?

Reply d. Thank you for this interesting comment. The BMQ has been formally translated to numerous languages and has largely been validated. A meta-analysis published in 2013, which contained studies performed in 18 countries and included more than 24 different chronic diseases, showed an overall association between the necessity-concerns differential and adherence (<https://www.ncbi.nlm.nih.gov/pmc/articles/PMC3846635/pdf/pone.0080633.pdf>). In our study, the survey of the official language of each participating country was used.

The Methods section (page 7) has been amended and reads: 'The survey was translated into the official language of the participating countries. Validated versions of the translated BMQ-general and BMQ-specific subscales were used when they were available^{22, 24-29}. Otherwise, translation from English and back translations were executed by two independent translators'

Cost and treatment tradition is likely to be different across regions. The use of, for example, newer medicines or the use of products with brand names vs generics varies across countries. However, our study only includes European countries, which means that European Regulatory Bodies (e.g. the European Medicines Agency) ensure minimum standards in the quality of medicines at the European market. Also, all countries across Europe have minimum social measures to avoid excessive medicines costs for patients.

In addition, the BMQ-Specific scales were disease-specific rather than medicinal product specific, while the BMQ-General applies to general beliefs about medicines. This means that the risk of biased answers based on cost of individual medicines is not likely to be major concern in this study.

The use of, for example, herbal and alternative medicine might also vary across regions. However, a report about the use of complementary and alternative medicine (CAM) (https://cam-europe.eu/wp-content/uploads/2018/09/CAMbrella-WP4-part_1final.pdf), did not find clear differences in the use of CAM across different European countries, neither on the use of Over-The-Counter medicines or prescription medicines. Also, we did not find significant differences between medicated and non-medicated women by region (i.e. northern, western, and eastern Europe) (see Table 1 of the manuscript).

Involving patients in the development of questionnaires is of high value. It is therefore common practice to involve them in the development. For instance, over 500 patients were involved in the development of the BMQ (<https://www.tandfonline.com/doi/pdf/10.1080/08870449908407311?needAccess=true>).

We now mention the importance of involving patients for future tools development in the Discussion (page 14), and it reads as follows: *'Future research should develop validated measures of pregnancy-specific beliefs about prescribed and over-the-counter medicines, and use patient-centered approaches in the development of the items.'*

e) It has been long acknowledged that the extremes of a Likert-type response tend to get less use than the more central choices causing an "anchor effect". Could this be a potential explanation for the results being independent of the underlying disease (except epilepsy where documentation for harm of medications is very strong)? Therefore, the intervals near the extremes may be further apart, than those near the center. This, by itself, disqualifies a Likert-type response as interval. See: <https://www.ncbi.nlm.nih.gov/pmc/articles/PMC4833473/>

Reply e. Anchor effects are known effects of Likert-scale questions. However, we do not think that this has affected our results (see our more detailed reply to your question 'b)'). Also, we used the same approach in the analyses as done by the BMQ developer, Professor R Horne, who groups the beliefs by BMQ-subscale and not by specific question.

f) The authors state in the paragraph starting at line 53 on page 11 that this study supports treatment adherence, although adherence was not specifically addressed in the survey. I guess that BMQ scores (and in particular necessity-concerns) are used as a proxy for adherence in studies.

Reply f. Indeed, this study does not measure adherence but medication use for chronic disorders in pregnancy. This is not a proxy for adherence since we do not know to what extent women were taking their medication as prescribed, even though they continued treatment in pregnancy. We have now made this passage clearer in the Discussion (page 11), which reads: *"Our study supports this association from a broader medication-taking perspective, and adds to this knowledge that the differences in beliefs between medicated and non-medicated pregnant women were shown across various diseases."*

As the reviewer can read, the second part of the revised sentence describes what type of knowledge the study adds (i.e. beliefs in medicated versus non-medicated women).

g) How many open-ended answers were used to create the five themes? Should this number be included in Results, and the respective text below Table 2?

Reply g. We have added the number of responders to the open-ended question to the results section (page 9).

The paragraph now reads: *“There were 359 women out of the 1219 who provided an open-ended answer to the question ‘Do you have any other comments about your medication use during pregnancy?’. From those answers, five themes were identified, that is confident, under-informed, confused, guilt, and avoiding medication. In all themes, there were women from the medicated and non-medicated groups. Examples of statements supporting these themes are presented Table 2.”*

Finally, due to the above mentioned challenges with measuring different chronic diseases, pregnant and beliefs of medications, more studies are needed to find the appropriate and more specific interventions. This notion should preferently be added to the conclusion.

Reply. We have added information about this to the Discussion section (page 13) and we hope we sufficiently addressed your valuable questions. The Discussion now reads: *“Innovative decision support tools that empower women and enable a shared decision-making approach about medications in pregnancy have been developed for some diseases ⁴⁵ and could lead to better-informed treatment decisions. Further development of such tools, also for other diseases, should be the focus of the future.”*